Potential urinary aging markers of 20-month-old rats

Li Xundou 1
Gao Youhe 2 gaoyouhe@bnu.edu.cn
1 Department of Pathophysiology, Institute of Basic Medical Sciences , Beijing , China
2 Department of Biochemistry and Molecular Biology, Beijing Normal University, Gene Engineering and Biotechnology Beijing Key Laboratory , Beijing , China
Ng Tzi Bun
Electronic publication date: 2016 Jun 7
Publication date: 2016
Volume: 4
Electronic Location ID: e2058
Received 2015 Nov 25; Accepted 2016 Apr 29
Copyright: ©2016 Li and Gao
Copyright year: 2016
Copyright holder: Li and Gao
License: This is an open access article distributed under the terms of the Creative Commons Attribution License, which permits unrestricted use, distribution, reproduction and adaptation in any medium and for any purpose provided that it is properly attributed. For attribution, the original author(s), title, publication source (PeerJ) and either DOI or URL of the article must be cited.
License URL: https://creativecommons.org/licenses/by/4.0/

Keywords: Urinary proteome, Aging

Funding: National Basic Research Program of China 2013CB530805 2012CB517606 2014CBA02005 2013FY114100 This work was supported by the National Basic Research Program of China (2013CB530805; 2012CB517606; 2014CBA02005 and 2013FY114100). The funders had no role in study design, data collection and analysis, decision to publish, or preparation of the manuscript.

==============================
Urine is a very good source for biomarker discovery because it accumulates changes in the body. However, a major challenge in urinary biomarker discovery is the fact that the urinary proteome is influenced by various elements. To circumvent these problems, simpler systems, such as animal models, can be used to establish associations between physiological or pathological conditions and alterations in the urinary proteome. In this study, the urinary proteomes of young (two months old) and old rats (20 months old; nine in each group) were analyzed using LC-MS/MS and quantified using the Progenesis LC-MS software. A total of 371 proteins were identified, 194 of which were shared between the young and old rats. Based on criteria of a fold change ≥2, P < 0.05 and identification in each rat of the high-abundance group, 33 proteins were found to be changed (15 increased and 18 decreased in old rats). By adding a more stringent standard (protein spectral counts from every rat in the higher group greater than those in the lower group), eight proteins showed consistent changes in all rats of the groups; two of these proteins are also altered in the urinary proteome of aging humans. However, no shared proteins between our results and the previous aging plasma proteome were identified. Twenty of the 33 (60%) altered proteins have been reported to be disease biomarkers, suggesting that aging may share similar urinary changes with some diseases. The 33 proteins corresponded to 28 human orthologs which, according to the Human Protein Atlas, are strongly expressed in the kidney, intestine, cerebellum and lung. Therefore, the urinary proteome may reflect aging conditions in these organs.

Introduction

A biomarker demonstrates a measurable change associated with a physiological or pathophysiological process. Urine can accumulate a lot of changes in the body, making it a good biological source for disease biomarker discovery (Gao, 2013; Gao, 2014a). As summarized in a recent paper (Wu & Gao, 2015), many physiological changes can be reflected in urine; thus, we have reason to expect that urine can be used to detect small, early changes in pathological and/or pharmacological conditions (Gao, 2015). Indeed, urine can be more sensitive for detecting changes than blood (Li et al., 2014), and several potential biomarkers performed better in urine than in blood (Gao, 2014b; Huang et al., 2012; Wu et al., 2013). Therefore, urine is an ideal source for biomarker discovery, and many urinary biomarkers have been reported in various types of diseases (Shao et al., 2011).

The major challenge in urinary biomarker discovery is that the urinary proteome can be affected by a variety of factors, such as gender (Guo et al., 2015), age, medications (Li, Zhao & Gao, 2014; Zhao et al., 2015), exercise (Kohler et al., 2009), and smoking (Airoldi et al., 2009), in addition to many other physiological variables and environmental factors. Thus, to identify specific urinary associations with a particular disease in clinical samples, human urinary biomarker studies must eliminate or balance these factors as much as possible by utilization of a large sample size. As changes in stable components of the urine proteome are more likely to serve as disease biomarkers (Sun et al., 2009), a better understanding of the effect of physiological and environmental factors on the urine proteome will facilitate the identification of urinary biomarkers. To achieve this objective, it is essential to introduce much simpler and more controllable systems, such as animal models. Because animal models have similar genetic backgrounds and the same living environment, it is easy to use a small sample size to establish associations between physiological or pathological conditions and the corresponding changes in urine. Therefore, screening potential disease biomarkers in animal models followed by validation in human samples may be a good strategy for urinary biomarker discovery.

Aging is a complex physiological process that induces a decline in the function in multiple organs. As urine is a window into the body, the urinary proteome should reflect pathophysiological changes and aging conditions in many organs. For example, urinary age-related peptide excretion patterns may allow the non-invasive detection of renal disease and demonstrate strong associations between human aging and human chronic kidney diseases (Zurbig et al., 2009). Urinary proteome changes in the elderly appear to reflect the physiological processes of aging and are particularly clearly represented in the circulatory and immune systems (Bakun et al., 2014). Therefore, a detailed analysis of the urinary proteome may be informative with regard to the physiological changes associated with aging.

Urine proteomics is a non-invasive, reproducible method that is easy to repeat at high frequencies. In the current study, the urinary proteomes of young (two months old) and old (20 months old, n = 9) rats were analyzed using LC-MS/MS. We identified age-associated urinary proteins and found that the urinary proteome may reflect aging conditions in many organs.

Materials and Methods

Animals and ethics statement

Pathogen-free, male Sprague-Dawley rats were purchased from the Institute of Laboratory Animal Science, Chinese Academy of Medical Science & Peking Union Medical College (Beijing, China). All animals were fed a standard laboratory diet and kept under controlled temperature (22 ± 1 °C) and humidity (65–70%). The study was performed after the rats had been allowed to acclimate for 1 week. This study was approved by the Institute of Basic Medical Sciences Animal Ethics Committee, Peking Union Medical College (Animal Welfare Assurance Number: ACUC-A02-2013-015). All rats received humane care in compliance with the institutional animal care guidelines approved by the Institutional Animal Care and Use Committee of the Peking Union Medical College.

Sample collection and preparation

Nine young rats (two months of age) and nine old rats (20 months of age) were allowed to acclimate for one week. The rat’s sexual maturity period (puberty) is at two months old, which is widely used in experiments. This may be equivalent to a 16-year-old human; and a 20-month-old rat is approximately equivalent to 55 years in a human (Andreollo et al., 2012). The rats were all weighed, and urine samples were collected using metabolic cages for 24 h. Urinary protein and creatinine concentrations were measured at the Peking Union Medical College Hospital. The urine samples were centrifuged at 5,000 × g for 30 min, and the supernatants were precipitated with 75% v/v acetone for 12 h followed by centrifugation at 12,000 × g for 30 min. After removing the supernatant, the pellets were thoroughly air-dried, resuspended in lysis buffer (8 M urea, 2 M thiourea, 50 mM Tris and 25 mM DTT) and subjected to protein quantitation using the Bradford assay. Proteins were digested using trypsin (Trypsin Gold, Mass Spec Grade, Promega, Fitchburg, WI, USA) with filter-aided sample preparation methods (Wisniewski, Zougman & Mann, 2009). Briefly, urinary proteins were loaded onto the filter unit (Pall, Port Washington, NY, USA), denatured at 50 °C for 1 h by the addition of 20 mM DTT and alkylated in the dark for 40 min by the addition of 50 mM IAA. Proteins were digested using trypsin (1:50) at 37 °C overnight. The digested peptides were desalted using Oasis HLB cartridges (Waters, Milford, MA, USA).

LC-MS/MS

The resulting peptides were analyzed by nanoLC-MS/MS with an Agilent 1200 HPLC system coupled to an LTQ-Orbitrap Velos mass spectrometer (Thermo Fisher Scientific, Bremen, Germany). Each sample was loaded, with a maximal volume of 8 µL, onto a Michrom Peptide Captrap column (MW 0.5–50 kD, 0.5 × 2 mm; Michrom Bioresources, Billeria, MA, USA) with a flow rate of 20 µL/min in 0.1% formic acid 99.9% water. The trap column effluent was transferred to a reverse-phase microcapillary column (0.1 × 150 mm, packed with Magic C18, 3 µm, 200 Å; Michrom Bioresources) in an Agilent 1200 HPLC system. The elution gradient for the reverse column changed from 95% mobile phase A (0.1% formic acid, 99.9% water) to 40% mobile phase B (0.1% formic acid, 99.9% acetonitrile) over 60 min at a flow rate of 500 nL/min. The LTQ-Orbitrap Velos was operated in data-dependent acquisition mode. Survey scan mass spectra were acquired by the Orbitrap in the 300–2,000 m∕z range, with the resolution set to a value of 60,000. The 20 most intense ions per survey scan were selected for CID fragmentation, and the resulting fragments were analyzed using the LTQ. Dynamic exclusion was employed with a 60-second window to prevent repetitive selection of the same peptide.

Database searching and protein quantification

The Mascot Daemon software (version 2.4.0, Matrix Science, London, UK) was used to search the MS/MS data against the SwissProt_rat database (release 2013_07; taxonomy: Rattus; containing 9,354 sequences). The carbamidomethylation of cysteines was set as a fixed modification. The oxidation of methionine and protein N-terminal acetylation were set as variable modifications. The specificity of trypsin digestion was set for cleavage after K or R, and two missed trypsin cleavage sites were allowed. The mass tolerances in MS and MS/MS were set to 10 ppm and 0.5 Da, respectively. Peptide and protein identifications were validated using Scaffold (version 4.0.1, Proteome Software Inc., Portland, OR, USA). Peptide identifications were accepted if they were detected with ≥95.0% probability by the Scaffold local false discovery rate algorithm, and protein identifications were accepted if they were detected with ≥99.0% probability and contained at least 2 identified peptides (Nesvizhskii et al., 2003). Label-free quantification was performed using the Progenesis LC-MS software (version 4.1, Nonlinear, Newcastle upon Tyne, UK), as previously described (Stoop et al., 2013).

Results and Discussion

The urinary protein-to-creatinine (U-pro/U-cr) ratio is an important index for monitoring kidney function. In this study, the U-pro/U-cr value increased approximately 2.2-fold with age (90 ± 15 in young rats versus 199 ± 60 in old rats, P < 0.001; Fig. 1), which suggests that kidney function is decreased in old rats.

Figure 1 Urine protein-to-creatinine ratio in young and old rats (P < 0.001, n = 9 per group).

Figure 2 Relative quantitation of 8 urine proteins identified as being related to aging (n = 9 pergroup; P < 0.05 for every protein).

Urinary proteome changes between young and old rats

To investigate how the urine proteome changes with age, 18 LC-MS/MS runs of urine samples in young and old rats (n = 9 per group) were performed using an LTQ-Orbitrap Velos. The false discovery rate (FDR) was adjusted to be less than 1%. A total of 371 proteins were identified, 194 of which were shared between the young and old rats. All proteins are listed in Table S1.

Based on label-free quantification using the Progenesis LC-MS software, 33 of the 371 proteins were altered (fold change ≥2, P < 0.05 and identified in each rat of the high-abundance group; Table S2). Among these, 15 proteins were increased and 18 decreased in old rats, suggesting that the urinary proteome changes greatly with age. Twenty of the 33 (60%) altered proteins have been reported as disease biomarkers (Shao et al., 2011), suggesting that there are high resemblances between aging and certain diseases. It is thus necessary to match the age of the disease and control groups in urinary biomarker research.

To find the most reliable age-related proteins, screening was performed with a strict standard: the protein spectral counts of all rats in the high-abundance group must be greater than those in the low-abundance group. Under these conditions, 8 proteins were found to be significantly changed in old rats and were thus considered age-associated proteins (Fig. 2); five of these proteins were increased in old rats, and 3 were decreased (Table S2). Two proteins, Ig gamma-2A chain C region and ceruloplasmin, also exhibit consistent trends in the urine of aging humans in previous study (Bakun et al., 2014), as aresult, it is proposed that these proteins are more reliable rat aging related proteins and suggest their human orthologs may be human aging related proteins. Therefore, screening potential disease biomarkers in animal models and then validating them in human samples may be a good strategy for urinary biomarker discovery.

Figure 3 Tissue distribution of the human orthologs of aging-associated rat proteins.

X-axis, human tissues; Y-axis, the number of strongly expressed proteins in human tissues compared the human orthologs using The Human Protein Atlas.

Human orthologs of rat proteins significantly affected by aging

It is assumed that orthologs (co-orthologs) retain similar functions among species (Koonin, 2005). We thus identified human orthologs of the proteins significantly altered with aging in rats. According to the 122.R_norvegicus.orthologues database and Ensembl Compare database (Shaye & Greenwald, 2011), 25 of the 33 rat urinary proteins correspond to 28 human orthologs (Table S2). By comparing the proteins with the human core urinary proteome, we further found that 16 human orthologs are relatively stable proteins in the normal human urinary proteome (Nagaraj & Mann, 2011; Sun et al., 2009). Therefore, because significant qualitative or quantitative changes to these stable proteins may suggest pathophysiological conditions, they could serve as potential urinary biomarkers (Sun et al., 2009).

According to the Human Protein Atlas, the significantly altered proteins are strongly expressed in the kidney, intestine, cerebellum and lung (Fig. 3); thus, the urinary proteome may preferentially reflect the aging conditions of these organs. By improving the depth of urinary protein identification in the future, the status of additional organs may be reflected in the urinary proteome (Fig. S1).

Supplemental Information

Table S1 All the proteins identified and quantified in this experiment.

Click here for additional data file.

Table S2 The 79 proteins changed between young and old rats with P < 0.05.

Click here for additional data file.

Figure S1 All tissues in which aging-associated rat proteins are distributed

Click here for additional data file.

Additional Information and Declarations

Competing Interests

Author Contributions

Animal Ethics

Data Availability

The authors declare there are no competing interests.

Xundou Li performed the experiments, analyzed the data, contributed reagents/materials/analysis tools, wrote the paper, prepared figures and/or tables, reviewed drafts of the paper.

Youhe Gao conceived and designed the experiments, reviewed drafts of the paper.

The following information was supplied relating to ethical approvals (i.e., approving body and any reference numbers):

This study was approved by the Institute of Basic Medical Sciences Animal Ethics Committee, Peking Union Medical College (Animal Welfare Assurance Number: ACUC-A02-2013-015).

The following information was supplied regarding data availability:

The raw data is available at https://figshare.com/s/d506c20578368a100b8f.

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
