# Peer review of "Potential urinary aging markers of 20-month-old rats"

_PeerJ, doi:10.7717/peerj.2058_

## Round 0.1 · original submission · Major Revisions

Dear authors

A major revision of your manuscript is required before it can be reconsidered for publication.Please make a point to point reply to each of the reviewers comments when you submit your revised manuscript. Highlight the parts which have been revised and refer to these modifications in your response letter.

Reviewer 1 ·

Basic reporting

1. Figure legends/titles are to be checked for English.
2. Sufficient description must be provided to the supplementary files.
3. Fig. 3 and corresponding places in text, criteria of "high" expression in human tissues according to ATLAS project are to be explained
4. A rationale of forming short list of 8 top proteins out of 33 FDR-passed entries is to be explained in more detail

Experimental design

I have no objections

Validity of the findings

I have no objections

Additional comments

Ocerall, compact manuscript with well formulated major message. Solving the issues mentioned above will hopefully improve the manuscript accessibility to the readership

Reviewer 2 ·

Basic reporting

The author did not show the significant of study.
No hypothesis has been mentioned.

Experimental design

The proteomics data should be validated by Western blotting analysis.
What's the rationale of selection of 2- and 20-month old rat in this study? are they clinical related?

Validity of the findings

Author mentioned that "Two proteins, Ig gamma-2A chain C region and ceruloplasmin, also exhibit 138 consistent trends in the urine of aging humans(Bakun et al. 2014)"; However, I cannot find these two proteins mentioned in this paper (refers to the following links).

http://www.ncbi.nlm.nih.gov/pmc/articles/PMC3889913/table/Tab2/
http://www.ncbi.nlm.nih.gov/pmc/articles/PMC3889913/figure/Fig2/

·

Basic reporting

The introduction and background do demonstrate how the research fills a knowledge gap, and what is the relevance of the approach proposed for the identification of biomarkers of ageing. However, the wording is rather muddled and demonstates a lack of basic understanding of physiological principles. Thus, for instance, proteins are not 'expressed' , or 'upregulated' in urine. Proteins are expressed in cells and tissues, then metabolized and broking down, while part of the proteins and metabolites are secreted in blood, and from the blood excreted in urine. Thus, by collecting all urine produced over a certain period of time (it is not stated in the article how long a period was chosen), e.g. 24 hours, information can be gathered on the proteins and metabolites produced in the body, and their amounts can be compared. However, it should be emphasised that the excretion of proteins strongly depends on the function of the kidney itself. These mechanisms are made insufficiently clear in the text, which could be much improved. The text should state more clearly why this approach is of use for identifying biomarkers of ageing: indeed, using animals with similar genetic backgrounds and the same living environment and comparing young versus old, may help to identify changes that are specific fo ageing (and not confounded by other factors) as a physiological process. The article conforms to the templates; the submission is self-contained.
Figures and tables are relevant and well-produced, and the raw data are made available as supplementary files.

Experimental design

First a disclaimer in this section: I dod not feel qualified to be able to evaluate the technical standards accoring to which the laboratory analyses have been carried out, as I am not a laboratory researcher.
This having been said, the submission describes original primary research within scope. The research question can be inferred from the text, but could have been stated much more clearly (see comments in section above).
It seems to me that sufficient information is provided to be able to reproduce the findings.

Validity of the findings

For a disclaimer see above.
The data are made available as supplementary files as spreadsheets and figures, which seems appropiate.
Conclusions are appropiately stated and connected to the original question. Speculation is clearly indicated as such.
A few points of criticism:
It is misleading to state that 'age-specific' proteins were identified. Only a comparison between old and young animals was made, not including a large range of ages.
Only 'shared' proteins between old and young are reported in the text and commented upon. What about the 'unshared' proteins. These might provide information that could potentially be even more interesting than the fold changes between shared proteins.

Additional comments

This is an interesting approach to identfy biomarkers of ageing. However, the wording could be much improved, and demonstates a lack of basic understanding of physiological principles. Thus, for instance, proteins are not 'expressed' , or 'upregulated' in urine. Proteins are expressed in cells and tissues, then metabolized and broking down, while part of the proteins and metabolites are secreted in blood, and from the blood excreted in urine. Thus, by collecting all urine produced over a certain period of time (it is not stated in the article how long a period was chosen), e.g. 24 hours, information can be gathered on the proteins and metabolites produced in the body, and their amounts can be compared. However, it should be emphasised that the excretion of proteins strongly depends on the function of the kidney itself. These mechanisms are made insufficiently clear in the text, which could be much improved. The text should state more clearly why this approach is of use for identifying biomarkers of ageing: indeed, using animals with similar genetic backgrounds and the same living environment and comparing young versus old, may help to identify changes that are specific fo ageing (and not confounded by other factors) as a physiological process.
A few points of criticism:
It is misleading to state that 'age-specific' proteins were identified. Only a comparison between old and young animals was made, not including a large range of ages.
Only 'shared' proteins between old and young are reported in the text and commented upon. What about the 'unshared' proteins. These might provide information that could potentially be even more interesting than the fold changes between shared proteins.

---

## Round 0.2 · Major Revisions

Please pay attention to the reviewers' comments.One of them is still not satisfied with the manuscript. Please further revise the manuscript.

Reviewer 1 ·

Basic reporting

improved in the revised version

Experimental design

sufficiently documented

Validity of the findings

meets PeerJ criteria

Reviewer 2 ·

Basic reporting

(1) The author still did not show the significant of study.
(2) I understood that the project is not easy to generate the hypothesis by author.

Experimental design

(1) Thanks for providing the information about "How old is a rat in human years?", please add the information into the manuscripts.

(2) by the way, please also add the rationale of selection of 2- (may be equivalent to 16 years old in human) and 20- (approximately equivalent to 55 years old in human) month old rat in this study.

Validity of the findings

Author mentioned that "Two proteins, Ig gamma-2A chain C region and ceruloplasmin, also exhibit 138 consistent trends in the urine of aging humans(Bakun et al. 2014)"; However, I cannot find these two proteins mentioned in this paper (refers to the following links).
http://www.ncbi.nlm.nih.gov/pmc/articles/PMC3889913/table/Tab2/
http://www.ncbi.nlm.nih.gov/pmc/articles/PMC3889913/figure/Fig2/

Author used urine sample from SD-rat, but author try to make conclusion using human protein ONLY by expectation, not by experimental data. Authors are recommended to provide the human data, if they insist to provide the conclusion using human aging-related proteins.

Additional comments

Authors are recommended to amend manuscript according to reviewers' comments in order to enhance the scientific thinking and logical flow of the manuscripts. Please don't provide excuse and try to avoid the improvement.

---

## Round 0.3 · accepted · Accept

Reviewer 2 has evaluated your revision and is satisfied. Many thanks.

Reviewer 2 ·

Basic reporting

Accepted at revised version

Experimental design

Accepted at revised version

Validity of the findings

Accepted at revised version

Additional comments

Accepted at revised version